# Polybutylene Succinate Processing and Evaluation as a Micro Fibrous Graft for Tissue Engineering Applications

**DOI:** 10.3390/polym14214486

**Published:** 2022-10-23

**Authors:** Giovanni Carlo Miceli, Fabio Salvatore Palumbo, Francesco Paolo Bonomo, Massimiliano Zingales, Mariano Licciardi

**Affiliations:** 1Dipartimento di Scienze e Tecnologie Biologiche Chimiche e Farmaceutiche (STEBICEF), Università degli Studi di Palermo, 90123 Palermo, Italy; 2Advanced Technology Network Center (ATeN Center), Università degli Studi di Palermo, 90128 Palermo, Italy; 3Dipartimento di Ingegneria, Viale delle Scienze, Università degli Studi di Palermo, ed.8, 90128 Palermo, Italy

**Keywords:** poly (1,4-butylene succinate), electrospinning, biomaterials, vascular grafts, bile ducts, tissue engineering

## Abstract

A microfibrous tubular scaffold has been designed and fabricated by electrospinning using poly (1,4-butylene succinate) as biocompatible and biodegradable material. The scaffold morphology was optimized as a small diameter and micro-porous conduit, able to foster cell integration, adhesion, and growth while avoiding cell infiltration through the graft’s wall. Scaffold morphology and mechanical properties were explored and compared to those of native conduits. Scaffolds were then seeded with adult normal human dermal fibroblasts to evaluate cytocompatibility in vitro. Haemolytic effect was evaluated upon incubation with diluted whole blood. The scaffold showed no delamination, and mechanical properties were in the physiological range for tubular conduits: elastic modulus (17.5 ± 1.6 MPa), ultimate tensile stress (3.95 ± 0.17 MPa), strain to failure (57 ± 4.5%) and suture retention force (2.65 ± 0.32 N). The shown degradation profile allows the graft to provide initial mechanical support and functionality while being colonized and then replaced by the host cells. This combination of features might represent a step toward future research on PBS as a biomaterial to produce scaffolds that provide structure and function over time and support host cell remodelling.

## 1. Introduction

The exploration of materials used to fabricate novel scaffolds for tissue engineering never stops. The scaffold is a tridimensional structure that should guide cell growth towards the reconstruction of tissues in terms of biological, functional, and morphological characteristics. It plays an essential role in tissue engineering providing a physiological like microenvironment that allows perfusion of nutrients and oxygen and transfers biochemical signals that influence cell phenotype and tissue formation [1]. Whether natural or synthesized, the scaffolds are typically biodegradable, biocompatible and can be used to release drugs and growth factors while providing mechanical support. When selecting a new biomaterial, whether synthetic or natural, the demands of clinical applicability, such as easy to acquire and long-term storage should be taken into consideration [2]. Synthetic polymers can be tailored on a molecular level to control degradation rate or modulate cell behaviour mimicking the ECM. They can also be blended with other polymers to fit any requirement that should be met to function as an integral part of the damaged tissue. The common materials for synthetic scaffolds are biodegradable polymers, such as polyvinyl alcohol (PVA), poly (lactic acid) (PLA), polyethylene oxide (PEO), poly (glycolic acid) (PGA), poly (ε-caprolactone) (PCL), polyethylene glycol (PEG) and polyurethane (PU) [3,4,5].

Over the past decades, several attempts have been made to prepare biodegradable polymeric materials with appropriate pore networks for different applications, like electrospinning (ES) thermally induced phase separation, rapid prototyping, and particulate leaching [6,7]. ES is a versatile technique highly used to develop fibrous scaffolds constituted by continuous fibers with diameters ranging from the nano to the microscale depending on flow rate, applied voltage, polymer concentration, and air gap between the spinneret and the fabrication target. The fibers obtained with this technique could replicate the approximative diameter of extracellular matrix components and even the orientations by adjusting the rotation speed of the collecting mandrel [8]. Nanofibrous architecture highly favorizes cell binding and other cell behaviour activities [9]. 

Several polymers either natural (e.g., collagen, silk fibroin, chitosan, alginate) or synthetized (e.g., PU, PCL, PLA, PGS) were successfully employed in fabricating porous scaffolds by electrospinning [10,11,12,13,14]. Although several electrospun conduits have been studied for different tissue engineering applications, none of them is currently used for small diameter conduits. Currently used prosthetic grafts (ePTFE, Teflon^®^ or PET, Dacron^®^) are satisfactory in large-diameter applications, but they fail to demonstrate long-term patency comparable to that of autologous grafts in small-diameter (<6 mm) settings due to their susceptibility to inflammation, thrombosis, intimal hyperplasia, and consequent compliance mismatch with the host vessels [15]. Furthermore, clinical demand for tubular scaffolds is not limited to vascular grafts. The global burden of cardiovascular diseases is on the rise and projected to affect 50% of American by 2030 [16]. The use of an artificial bile duct instead of choledochojejunostomy, which is the standard technique for biliary reconstruction, could maintain a physiological conduit for bile. Moreover, there are other needs for tubular conduct substitutes like peripherical limb revascularization, arteriovenous fistulae for haemodialysis and peripherical nerve regeneration. In recent years, for these reasons, several research groups are studying other tubular scaffolds substitutes [17,18,19]. These grafts should be permanently implanted to restore the effectiveness of a tract that is no longer able to correctly transport the fluids. Designing requirements are hemocompatibility, elasticity, avoid leakage, compatibility with the standard sutures, mechanically resistance, sterilizability, reliability and with the appropriate length and diameter. Furthermore, they must provide initial mechanical support and functionality, then are supposed to degrade and be replaced by the cells [2]. Their properties and degradation profile could be controlled during the manufacturing process to provide the best outcomes for ECM synthesis and tissue regeneration.

In this study, we introduce the use of Poly(1,4-butylene succinate) (PBS) as an electrospun polymeric tubular scaffold for tissue engineering applications. PBS is a commercially available, biodegradable, thermoplastic aliphatic polyester, synthesized by the polycondensation of succinic acid and butanediol. These monomers can be either derived from fossil fuels or renewable resources and currently commercially available PBS is synthesized from chemically derived monomers. Butanediol (BDO) can be derived from glucose using a total biosynthetic route. Succinic acid is an important intermediate of the TCA cycle and is produced by several microorganisms as a fermentation product under anaerobic conditions. Therefore, complete bio-based PBS can be produced at an industrial scale [20]. 

PBS is insoluble in water, its melting temperature is 120 °C and the glass transition temperature is below 0 °C. It can be sterilized with several techniques without compromising the scaffold’s properties. Lastly, it has excellent processability and proven biocompatibility making it a promising polymer for various biomedical applications [21,22,23,24,25]. 

Cicero et al. used PBS as a planar microfibrillar scaffold implanted as a conduit, in a rat model, to preserve nerve continuity and promote its regeneration. They observed biodegradability from high resolution MRI investigation showing complete reabsorption in 120 days post implant [26]. Almeida et al. produced and characterized anisotropic planar scaffolds through weft knitting. Demonstrating that PBS is a promising material for tissue engineering applications due to the high level of processing control and because it supports cell adhesion and proliferation [27]. Di Prima et al. chose PBS as a polymeric ocular insert due to its manageability in the electrospinning process and hydrophobicity. They modified the electrospun scaffold surface via plasma-induced chemical functionalization to improve biomimetic and mucoadhesive properties [28]. 

The mechanical properties are strictly dependent on the presence of diisocyanates used as chain extenders. High molecular weight PBS synthesized without chain extenders shows a brittle behavior, with very short elongation at break, while the use of isocyanates significantly improves its elongation [29]. Therefore, we chose PBS extended with 1,6-diisocyanatohexane for use in a dynamic environment because it is capable of sustaining and recovering from short-term deformation making it suitable for these applications. Several works use electrospinning to produce planar micro or nanofibrous scaffolds from PBS, but none analyzed small-diameter conduits [20,21,22,24]. In this study, small diameter (2.6 mm) conduits were produced by electrospinning from PBS. Potential applications include vascular grafts, artificial bile ducts, peripherical limbs revascularization, arteriovenous fistulae for haemodialysis or peripherical nerve regeneration. Scaffold morphology and mechanical properties were exhaustively studied, and cytocompatibility was evaluated in vitro using adult normal human dermal fibroblasts. Moreover, the stability of the scaffold in physiological fluids, such as bile and plasma, resulted sufficient to provide initial mechanical support and functionality, while being colonized and then replaced by the host cells.

## 2. Materials and Methods

### 2.1. Materials

Poly (1,4-butylene succinate) extended with 1,6-diisocyanatohexane (T_m_ 120 °C) (PBS), 1,1,1,3,3,3-hexafluoroisopropanol (HFIP) and Dulbecco’s phosphate buffered saline (DPBS) were purchased from Aldrich Milan, Italy. This merchant specifies only the T_m_ for PBS, but Fabbri et al. performed a GPC analysis to evaluate the molecular weight (81.2 kDa) [21].

Dulbecco’s modified Eagle’s medium (DMEM), fetal bovine serum (FBS), trypsin, l-glutamine, penicillin, streptomycin, and amphotericin were purchased from Euroclone group (Milan, Italy).

Porcine bile was extracted from pigs at Istituto Zooprofilattico della Sicilia “A. Mirri,” Palermo, Italy accordingly with European rules on animal experiments.

Human blood was extracted from volunteers upon informed consent and isolated at the University of Palermo, Palermo, Italy.

NHDF-Ad-Human Dermal Fibroblasts, Adult were obtained from Lonza bioscience and used after 9 doublings. The cell line was grown in a minimum essential medium [Dulbecco’s modified Eagle’s medium (DMEM)] supplemented with 10% (*v*/*v*) fetal bovine serum (FBS), 2 mM l-glutamine, 100 um/mL penicillin, 100 μg/mL streptomycin, and 2.5 μg/mL amphotericin B (all reagents were from Euroclone, Milan, Italy) under standard conditions (95% relative humidity, 5% CO_2_, 37 °C).

### 2.2. Fabrication of the Fibrous Graft through Electrospinning

PBS was blended at 15% *w*/*v* in 1,1,1,3,3,3-hexafluoroisopropanol (HFIP) under mechanical stirring at room temperature. The polymer solution was collected by a precision 10 mL syringe and placed in an NF 103 Electrospinning (MECC, Fukuoka, Japan). The flow passed through a PTFE tube and then in a steel flat needle (22 gauge) with 15.5 cm gap between the needle tip and the collector. One high-voltage generator was employed with a positive voltage (+12.5 kV) to charge the steel capillary containing the polymer solution while the stainless-steel collector rod was maintained at ground voltage. Rotational speed and translational movement were kept constants for all the fabrications (40 rpm, 0.8 cm/s, span 5 cm). The humidity was maintained in a range between 23% and 27%. Deposited scaffolds were soaked in water for 10 min, then the electrospun grafts were quickly slipped through the metal rod and dried.

### 2.3. Morphological Characterization of the Electrospun Graft

In this study, scanning electron microscopy (SEM) was adopted as an investigation method to study the morphology of ES scaffolds. The samples were dehydrated and mounted on aluminium stubs with double adhesive carbon tape. Stubs were vacuum-coated with a 5-nm thick layer of gold (Sputter Coater LuxorAu, Luxor Tech, Nazareth BELGIUM) and observed by a Phenom PRO X SEM.

The fiber diameter, morphology and pore size were evaluated using the ImageJ (1.52Q Wayne Rasband National Institute of Health, Bethesda, MD, USA) software by measuring the diameters of at least 50 fibers and hundreds of pores from each SEM micrograph.

A micro-Computed Tomography (𝜇CT) scanner (Skyscan 1272, Bruker Kontich, Belgium) has been used to run the 𝜇CT analysis. One side of the sample was held fixed on the instrument rotating support by laying a layer of wax. The images were acquired by setting a rotation step of 0.2° with a pixel size of 3 um. An X-ray beam with a source voltage of 40 kV and a source current of 250 uA was used. To image the whole sample in high resolution mode the functions ’oversize’ and ’batch Scanning’ were used. These functions allow to scan the whole object by dividing it into sub-objects and then reconstructing the sample. In this case, the uCT of the full cylinder has been achieved by combining 5 scans of sub-cylinders.

### 2.4. Water Permeability and Wettability Evaluations of the Graft

The water vapour transmission rate (WVTR) was evaluated, with distilled water, at 25 °C and 38 °C by using a PermeH2O ExtraSolution instrument at 37 °C and with a relative humidity of 50%. The analysis was performed in triplicate and results were expressed as mean permeability (g/m^2^·24 h) ± standard deviation.

The porosity was calculated as a difference between the predicted scaffold density and the polymer density (gravimetry) by the Equation (1) as follows:(1)$$P=\left(1-\frac{M}{v\times \rho \times 1000\;}\right)\times 100$$
where *P* (%) is porosity, *M* is the weight of the graft [g], *v* is the volume of the graft [m^3^], *ρ* is the density of PBS [g/cm^3^].

The hydrophilicity of the grafts was studied by a video contact angle instrument (FTA 1000 C class, First Ten Angstroms, Portsmouth, VA, USA). Deionized water was deposited on the graft surface and an image of the drop was recorded. From the images of the droplets on the surface of the mats, at scheduled time intervals up to 30 min, acquired by a digital camera and processed by a software program, the average values of the water contact angles were determined. The contact angle experiments were performed in triplicate at room temperature, and the results were expressed as mean value ± standard deviation.

### 2.5. Degradation Evaluation

The degradation test was carried out at 37 °C in three different buffer solutions sodium azide/PBS 0.02 M pH 7.4, porcine bile and human plasma. The electrospun grafts were cut, washed in distilled water, lyophilized, and weighed (w_0_). The weight loss was measured at scheduled time intervals, up to one month, after washing them four times in ultra-pure water and lyophilising them. These experiments were performed in triplicate, and results were expressed as mean value of recovered weight ± standard deviation. Results were plotted against t_0_. Finally, the samples were coated with gold and observed with SEM.

### 2.6. Uniaxial and Suture Retention Tests

Mechanical testing was performed, by tensile measurements, on the fibrous scaffolds using a Bose TA Instruments ElectroForce Test Bench System. Dry scaffolds were cut into strips, width = 5 mm and length =30 mm, along their circumferential or longitudinal directions. The thickness measurement of each specimen, obtained with a digital caliber, was 400 um. The initial length between the clamps was 15 mm and firm retention of the scaffold was ensured by the rugged metal clamps of the tensile system. The dry specimens from each direction were tested at room temperature by pulling at 10 mm/min crosshead speed until rupture following 10 cycles of preconditioning to 15% strain. Load–displacement curves were computed to obtain stress–strain relationships according to current length and cross-sectional area with the assumption of incompressibility.

Ultimate tensile stress (UTS) and strain to failure (STF) were considered, respectively as the maximum stress value before failure and its corresponding value of strain. Elastic modulus was estimated by calculating the slope of the stress-strain curves in the elastic region. Suture retention tests were performed on rectangular specimens clamped at the edge located opposite the suture. The thread was passed through the material using the provided triangular needle (2–0) and closed into a loop by multiple knots. The specimen had a free length of 20 mm, a width of 11 mm and 0.2 mm thickness. The suture bite was centred to the specimen width and its distance from the clamp was 18 mm. The suture loop was first pulled at 0.2 N while the specimen was held fixed. Once the suture wire was taut, a pulling rate of 1 mm/s was applied until final specimen failure, characterized by suture pullout.

### 2.7. In Vitro Studies

The electrospun grafts were cut into square shape of 1.5 mm × 1.5 mm and sterilized with gas plasma for in vitro cell culture studies. Then, were mounted on sterile CellCrown inserts (Scaffdex) and transferred into a 48-wells tissue culture polystyrene plate. Cultured NHDF were suspended in complete DMEM, about 30,000 cells per well, and used for culture seeding. Cell morphology and proliferation were studied through SEM and immunocytochemistry to visualize cytoskeleton organization and count cell nuclei. At each timepoint, the samples were washed with DPBS and fixated in 4% formaldehyde in PBS for 15 min at room temperature. Nuclei were counterstained with 4–6-diamidino-2-phenylindole (DAPI) for 10 min after 24, and 72 h in culture. Then, the samples were washed three times with PBS for 5 min, mounted onto glass slides and viewed under a Zeiss Axio vert. A1.

For the proliferation and attachment assessment, the number of DAPI-stained nuclei on the surface of the scaffold was counted using the image editing software ImageJ.

After the fluorescence analysis, the samples were washed four times in ultra-pure water and dehydrated using gradual ethanol concentrations (30%, 60%, 90% *v*/*v* and pure ethanol) for 10 min each. Finally, the samples were treated with hexamethyldisilazane (HMDS) and dried under a flow hood. The samples were coated with gold and observed with SEM.

Three millilitres of blood from a healthy donor were collected in Vacuette^®^ containing sodium edetate. The anticoagulated blood was kept for 60 min at 37 °C, after which the tubes were centrifuged at 500× *g* for 5 min. The supernatant was removed, and the pellet washed 5 times with 2.5 mL of PBS. It was subsequently diluted 1:50 in PBS providing the source of haemoglobin for haemolysis assay. The rate of haemolysis was evaluated by determining the relative amounts of haemoglobin released into solution phase from erythrocytes in diluted whole blood exposed to the test materials. Electrospun samples were cut into 1 cm^2^ squares and covered with the prepared solution. The negative controls contained only the erythrocytes solution in PBS and positive controls contained 2% Triton 100× in distilled water to induce maximal lysis of erythrocytes. The tested samples were kept for 60 min at 37 °C, after which the solutions were centrifuged at 500× *g* for 5 min. The supernatant containing the solubilized haemoglobin was removed and its absorbance was measured at a wavelength of 570 nm (three measurements from each sample). After the analysis, the samples were fixated in 4% formaldehyde in PBS for 15 min at room temperature. Then washed four times in ultra-pure water and dehydrated using gradual ethanol concentrations (30%, 60%, 90% *v*/*v* and pure ethanol) for 10 min each. Finally, the samples were coated with gold and observed with SEM.

## 3. Results and Discussion

### 3.1. Fabrication and Morphological Assessment of PBS Scaffolds

In this work, we developed and tested a tubular scaffold for tissue engineering applications obtained by electrospinning. The potential of polybutylene succinate to produce small diameter nanofibrous conduits is far from being fully explored. Therefore, we have focused our work on the optimization of the processing and characterization of the three-dimensional tubular construct. The setup and processing parameters used for the above goal are reported in Figure 1.

The presented electrospinning setup can produce tubular constructs ranging from 2.6 to 10 cm in diameter and 2 to 12 cm in length. This versatility allows us to tailor the dimensions of this scaffold to different applications.

Macroscopically, fiber deposition was smooth and homogeneous along the metal rod, without any gross defects (Figure 2A). An almost linear relation between the thickness of the electrospun layer and the deposition time was observed. The grafts were then plastically modified obtaining a corrugated geometry to minimize kinking when flexed (Figure 2B). Indeed, this geometry allows them to be flexed up to 84° without compromising the lumen and avoid local perturbations in the fluid dynamics.

The micro-CT scans were adopted as an investigation method to study the whole morphology and size of the graft shown in (Figure 2C) This analysis allowed us to inspect completely the graft structure and precisely measure wall thickness (mean size 490 ± 20 um) and inner diameter (mean size 3.62 ± 0.08 mm). The resulting sample reconstruction underlines the absence of delamination and homogeneous morphology.

SEM was adopted as an investigation method to study the morphology and orientation of nanofibers. Observed under SEM (Figure 3), the scaffolds appeared as a random fiber mat characterized by a homogeneous matrix with fiber diameters equal to 1.19 ± 0.13 um (as generated by fiber distribution diagram analysis).

A common challenge in tissue engineering is the difficulty of obtaining highly bulk-cellularized scaffolds due to intrinsic scaffold limitations such as insufficient pore size or interconnectivity. High porosity, pore interconnectivity, and large surface area to volume ratio of electrospun scaffolds dictate the extent of cellular infiltration and tissue ingrowth into the scaffold. Furthermore, they influence a range of cellular processes and are crucial for the diffusion of nutrients, metabolites, and waste products. We evaluated pore sizes using the ImageJ software, by measuring the area of thousands of pores and porosity was calculated as a difference between predicted scaffold density and the polymer density (gravimetry). The results showed an 88.6 ± 0.9 % porosity and a maximum pore size of 15.2 um (Table 1). The SEM analysis was also utilized to evaluate the potentially dangerous interaction with ethanol and gas plasma utilized to sterilize the scaffolds for in vitro experiments. The revealed quality of the microstructure demonstrated no morphological modifications.

### 3.2. Surface Properties Evaluation of the Electrospun Graft

The water vapour transmission rate (WVTR) of the graft was evaluated, both at 25 °C and 38 °C, according to the American Society for Testing and Materials (ASTM) standard.

The result showed a linear increase of WVTR between 0 and 600 s and then the value remained constant. The WVTR after 10 min are remarkable, 14,000 g/m^2^·24 h at 38 °C and 6500 g/m^2^·24 h at 25 °C (Figure 4A). As reported by Xu et al., a medium permeability membrane, WVTR 2000 g/m^2^·24 h, provide the optimal local environment to promote the proliferation and functions of fibroblasts [30]. 

The resulting value of WVTR is three times higher than their highest value of permeability membrane which was considered extremely high. This data is also confirmed by the contact angle dynamic measure (Figure 4B).

Indeed, the water contact angle of the electrospun graft started at 114° indicating a hydrophobic surface and then decreased gradually since the surface is permeable to water. Taken together, the data on the contact angle and the WVTR studies suggest that the PBS graft should allow the physiological diffusion of metabolites through the tubular wall (Table 2).

### 3.3. Stability in Physiological like Environments

The degradation test was performed in three different buffer solutions to mimic the physiological like environment of different human tubular conduits. Bile is the common fluid present inside the bile ducts, used to digest lipids. It is a basic solution composed of water, salts, and bilirubin. It was selected to evaluate the degradation of PBS graft as an artificial bile duct for biliary reconstruction.

Plasma is the intravascular part of extracellular fluid without blood cells. Indeed, it was used to evaluate the degradation of PBS as an artificial vascular substitute. Lastly, phosphate buffered saline was used to evaluate the hydrolytic resistance of PBS scaffolds. Ref. [21] The results, in Figure 5, showed no remarkable difference in the degradation of these grafts up to one month. This would likely help the host remodelling mechanisms to gradually replace the scaffold with native tissue while the scaffold provides mechanical integrity over an extended period.

### 3.4. Mechanical Properties

Mechanically, these PBS grafts showed a stress-strain curve, at room temperature and dry conditions, characterized by an initial linear region between 0 and 18% strain and a plastic region until rupture occurs. Indeed, these grafts retain their geometry and properties upon short-term mechanical conditioning when traction is applied longitudinally (Figure 6). It is crucial to withstand physiological pressures without experiencing permanent deformation or bursting.

The elastic modulus, obtained dividing young stress (YS) 3.1 ± 0.1 MPa by young strain (YN) 17.7 ± 1% in the elastic region, is 17.5 ± 1.6 MPa. Which is stiffer than the native soft tissue like other aliphatic polyesters such as poly (glycolic acid), poly (L-lactic acid), and their copolymers. Nevertheless, it could be adjusted to the desired range inducing anisotropy by regulating the mandrel rotation or designing a multilayer graft that combines the properties of different synthetic or natural materials. Both these solutions will have a significant implication on the mechanical properties giving us the possibility to tune them based on the application. The region between 17.7 ± 1 and 57 ± 4.5% strain is where strain hardening, caused by alignment of polymer chains in the direction of the load, prevailed. Indeed, this is the reason why stress increased, after yield stress, until necking occurred. The ultimate tensile strength, the highest engineering stress the material can endure, is 3.95 ± 0.17 MPa. Instead, strain to failure (STF) indicates how much the material has stretched when necking overtakes strain hardening 57 ± 4.5%. Necking is when one region of the sample becomes thinner than the rest concentrating the stress.

The Suture retention force was 2.5 ± 0.14 N when the force was applied on the longitudinal axe of the graft and 2.65 ± 0.32 N when the force was applied on the circumferential axe. This result is consistent with the SEM analysis result that underlines the random fiber orientation. Suture retention strength is crucial for an implanted graft to withstand the tension at the anastomosis. The generally accepted minimum requirement for the suture retention force for a small-diameter graft is 2 N. The SRF of the PBS graft prepared in this study was higher than this standard and then autologous substitutes like human saphenous vein (1.81 ± 0.02) and internal mammary arteries (1.4 ± 0.01). [31]

These characteristics, reassumed in Table 3, suggest that the PBS graft may represent an attractive biomaterial for tissue engineering applications. Furthermore, since it is possible to reproducibly control the anisotropy of electrospun PBS by varying the rotational speed of the mandrel, fiber orientation end consequent mechanical properties could be tailored to that of the vessel targeted for replacement.

### 3.5. In Vitro Studies

Cell’s interaction with the graft was evaluated in static conditions up to 72 h. The scaffold’s structure was designed to imitate the composition of the ECM to promote cell’s growth. Indeed, the results showed a proliferation of normal human fibroblasts after three days. The samples were washed several times to remove non-viable cells and then fixated in formalin and stained with DAPI to acquire reflectance images of the numbers of cells/mm^2^ (Figure 7A,C). Then these images were processed on ImageJ software to automatically count the nuclei spots. The results showed a remarkable increase in cell density that underlines the cytocompatibility of the produced PBS grafts. Since this technique gives us only an indication of the cell’s number, we processed the samples with hexamethyldisilazane to completely remove water. They were then analyzed with SEM to evaluate the cell’s morphology and their interactions with the fiber mats (Figure 7D,E). The dehydration process modified the cell’s morphology and removed several cells, but it gives us an overview of the cell’s orientation and interaction with the fiber mats confirming the cytocompatibility of the produced PBS grafts. Furthermore, both analyses showed no cells inside the graft’s wall or on the outer surface, confirming that this scaffold morphology act as a barrier for cell infiltration, without preventing its adhesion.

The presence of certain materials can negatively affect red blood cells and destroy their cell membrane causing the release of haemoglobin into the solution. [32] Haemolysis rate was measured using colorimetric assay of released haemoglobin from red blood cells. The lowest degree of haemolysis, absorbance of 0.042 ± 0.0012, was provoked by negative controls containing PBS with the addition of diluted whole blood. The highest was provoked by positive control group, 2% Triton X 100 in pure water, where the absorbance of released haemoglobin reached the value of 0.24 ± 0.05. The absorbance of released haemoglobin provoked by the electrospun sample is 0.045 ± 0.00073, which is very similar to the negative control (Figure 8A). Blood cells, attached to the fiber mat are shown in Figure 8B.

The degree of haemolysis (%) can be expressed according to the following formula:$$\%\;\mathrm{haemolysis}=\frac{\left(Ts-Tn\right)}{\left(Tp-Tn\right)}\times \;100$$
where *Ts* is the average absorbance of the test sample group, *Tn* is the absorbance of negative control group and *Tp* is the absorbance of positive control group. The degree of haemolysis of blood contacting materials must be <5% according to ISO 10993-4:2002 standard and ASTM F756-00(2000). Therefore, our electrospun PBS is considered as non-haemolytic material since the resulting value is 1.8 ± 0.4%.

## 4. Conclusions

In the present study, a tubular scaffold of electrospun PBS for tissue engineering applications was described. The main advantages of this aliphatic polyester are regular and controllable structure, robust mechanical properties, and a slow degradation rate. Furthermore, it can be mass-produced, shipped, and stored in a reproducible and controlled way and at an economically viable cost.

The selected processing technique has the advantage of tuning the ranges of lengths, diameters and even anisotropy. Therefore, it is possible to design different proprieties to fit the requirements for several tubular scaffold applications. Moreover, the fibers obtained by electrospinning confers to the structure of interconnective and tunable pores that, combined with the permeability of this material, allow the physiological exchange of metabolites and catabolites. A morphological and mechanical characterization showed that these grafts retain their geometry and properties upon short-term mechanical conditioning. The shown degradation profile allows the graft to provide initial mechanical support and functionality while being colonized and then replaced by the host cells. Furthermore, it is suitable for implantation since it is easy to be handled, sutured, non-haemolytic and has a remarkable kink resistance. This combination of features might represent a step toward future research on PBS as a biomaterial to produce scaffolds that provide structure and function over time and support host cell remodelling.

## Figures and Tables

**Figure 1 polymers-14-04486-f001:**
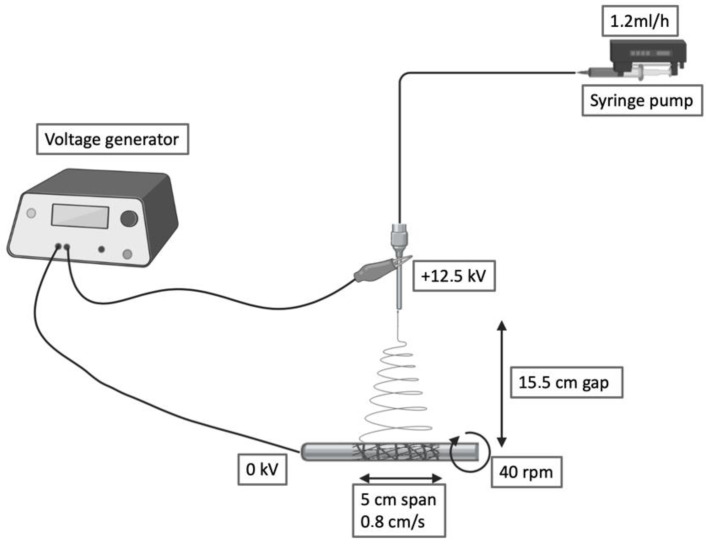
Schematic of the electrospinning setup and processing variables.

**Figure 2 polymers-14-04486-f002:**
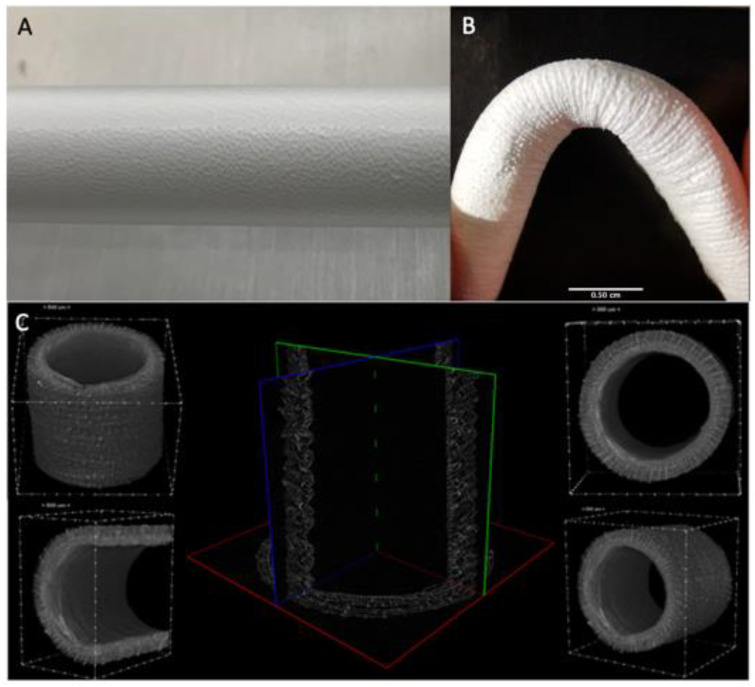
(**A**) Details of the ES deposition on the rotating collector (**B**) Kinking resistance of the graft at 84° (**C**) uCT analysis of the graft structure.

**Figure 3 polymers-14-04486-f003:**
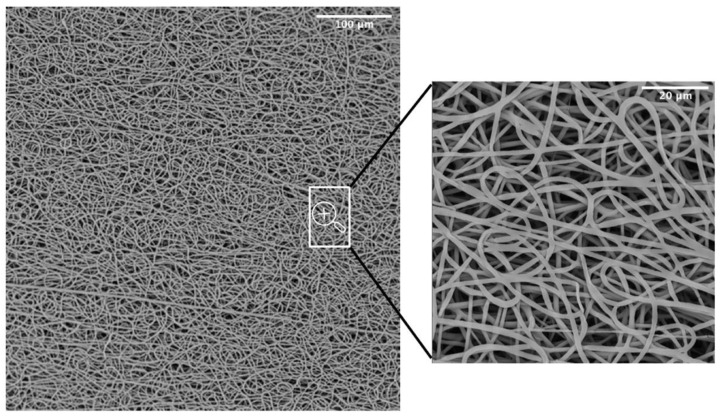
SEM micrograph of the PBS graft.

**Figure 4 polymers-14-04486-f004:**
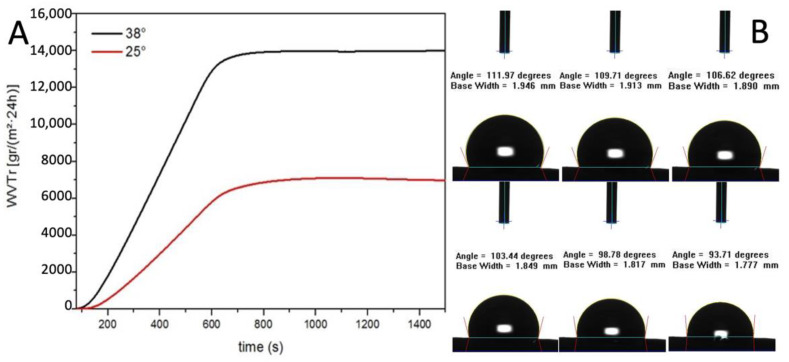
(**A**) Water vapour transmission rate evaluation (**B**) Contact angle measurements.

**Figure 5 polymers-14-04486-f005:**
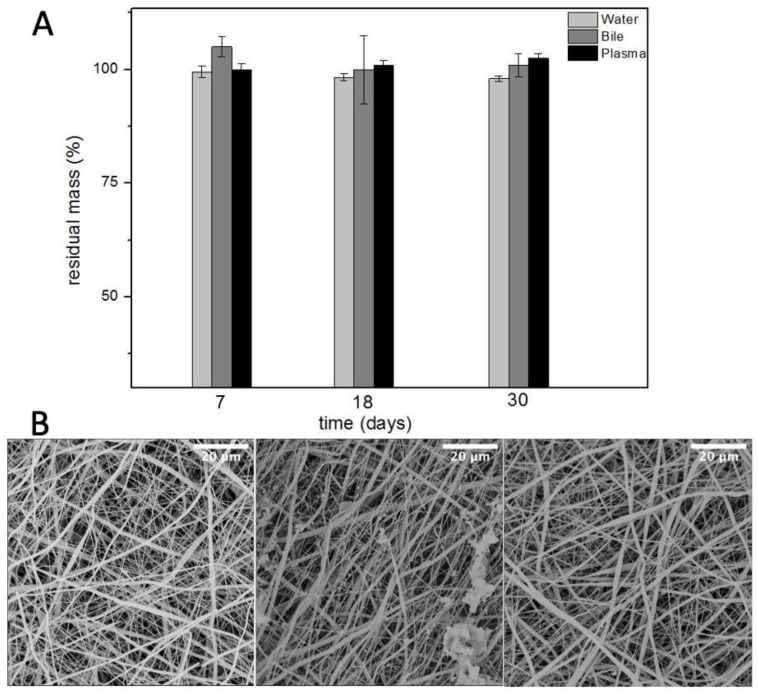
Residual mass (**A**) and SEM micrographs (**B**) of electrospun PBS after degradation in water (left), bile (centre), and plasma (right) (*n* = 5).

**Figure 6 polymers-14-04486-f006:**
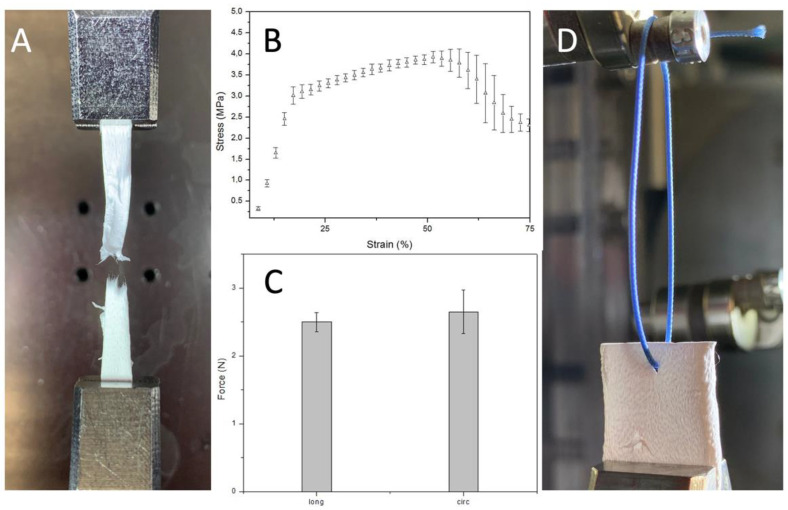
(**A**) sample detail after failure (**B**) Uniaxial stress-strain curve (*n* = 3) (**C**) Uniaxial suture retention force (*n* = 3) (**D**) Sample details of PBS graft before suture pullout.

**Figure 7 polymers-14-04486-f007:**
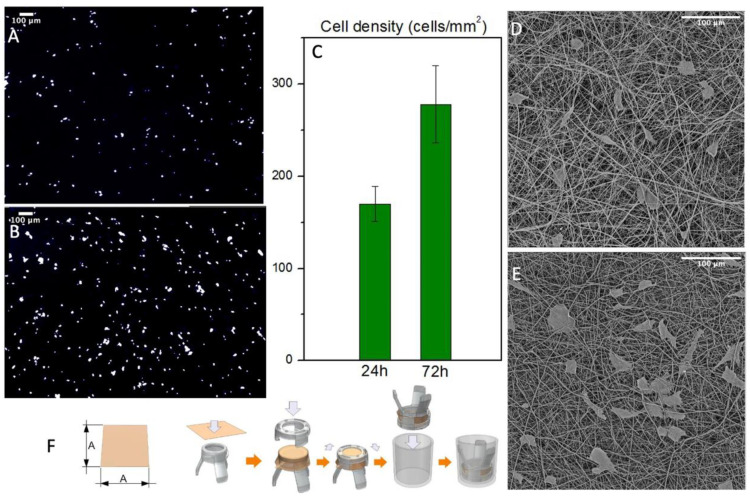
(**A**,**B**) NHDF nuclei when stained with DAPI after 24 and 72 h respectively (**C**) Cell density count with ImageJ of the stained nuclei (*n* = 3) (**D**,**E**) SEM micrographs of the cultured scaffold after 24 and 72 h respectively (**F**) Mounting process of the scaffolds on Scaffedex supports.

**Figure 8 polymers-14-04486-f008:**
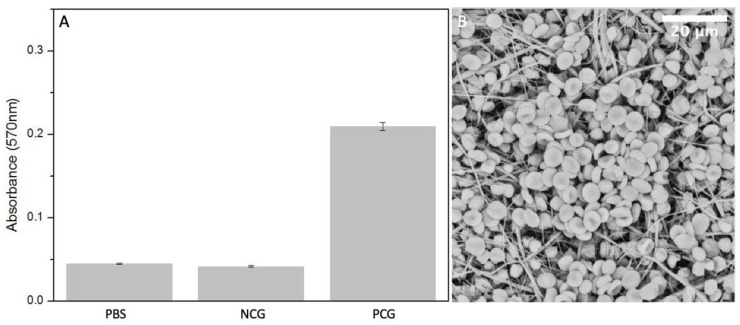
(**A**) Measured absorbance after haemolysis assay in positive control group PCG, negative control group NCG and electrospun PBS (*n* = 5). (**B**) SEM micrograph of electrospun PBS after incubation with diluted whole blood.

**Table 1 polymers-14-04486-t001:** Porosity, inner diameter, and wall thickness calculated from the uCT data, pore size and fiber diameter calculated from SEM and kink resistance.

	Porosity	Inner Diameter	Wall Thickness	Max Pore Size	Fiber Diameter	Kinking Resistance
PBS graft	88.6 ± 0.9%	2.62 ± 0.08 mm	490 ± 20 um	15.2 um	1.19 ± 0.13 um	84°

**Table 2 polymers-14-04486-t002:** Parameters obtained from WVTR and contact angle analysis.

	WVTR g/m^2^ 24 h 25°	WVTR g/m^2^ 24 h 38°	Contact Angle t = 0 s	Contact Angle t = 1800 s
PBS graft	6500	14,000	114°	93°

**Table 3 polymers-14-04486-t003:** Parameters obtained from uniaxial tensile test and suture retention test.

	E	UTS	STF	SRF	YS	YN
PBS grafts	17.5 ± 1.6 MPa	3.95 ± 0.17 MPa	57 ± 4.5%	2.5 ± 0.14 N	3.1 ± 0.1 MPa	17.7 ± 1%

## Data Availability

The data that support the findings of this study are available from the corresponding author upon reasonable request.

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
