# Peer review of "Polybutylene Succinate Processing and Evaluation as a Micro Fibrous Graft for Tissue Engineering Applications"

_polymers, 2022, doi:10.3390/polym14214486_

Round 1

Reviewer 1 Report

The authors report the development of polybutylene succinate electrospun nanofibrous tubular scaffolds for tissue engineering applications. This is an interesting work, but there are several issues that authors should consider:

-          The English language must be revised by a native speaker;

-          Figure 7 : improve quality of images A and B;

-          Mechanical properties:

o   Stress-strain and suture retention tests: How many samples were analyzed?

o   Radial dynamic compliance should be evaluated: I invite the authors to see the ISO 7198:2016 standard;

o   Lines 330 to 3369: the analysis is incorrect! The authors discuss the mechanical properties of a polymer as if it was a metal (dislocation interaction?????).

-          For the applications envisaged hemocompatibility and thrombogenicity are needed and should be presented;

-          Remodeling process time depends on the tissue: Please indicate the time scale for the remodeling of the tissues identified in the work (vascular grafts; bile ducts). Evaluation of the suitability of the scaffolds for tissue engineering purposes must take this time scale in consideration.

I believe that after amendments, considering the abovementioned issues, the manuscript may be published.

Author Response

Comments and Suggestions for Authors

The authors report the development of polybutylene succinate electrospun nanofibrous tubular scaffolds for tissue engineering applications. This is an interesting work, but there are several issues that authors should consider:

  1. The English language must be revised by a native speaker;

According with reviewer, the manuscript grammar is now updated.

  1. Figure 7: improve quality of images A and B;

The quality of images 7 A and B was improved.

  1. Stress-strain and suture retention tests: How many samples were analyzed?

Authors thank reviewer for the question. Accordingly this information was added in line 383 of revise manuscript.

  1. Radial dynamic compliance should be evaluated: I invite the authors to see the ISO 7198:2016 standard.

Authors thank reviewer for the question and agree that this analysis is in our plan for future works. Indeed, now we don’t have the instruments to perform it properly.

  1. Lines 330 to 369: the analysis is incorrect! The authors discuss the mechanical properties of a polymer as if it was a metal (dislocation interaction?????).

Authors thank he reviewer for noting this misunderstanding; accordingly the words “dislocation interaction” were deleted and sentence rewritten in line 403-405 of revised manuscript.

  1. For the applications envisaged hemocompatibility and thrombogenicity are needed and should be presented;

Authors thank he reviewer for the suggestion, and accordingly hemocompatibility test have been performed. The analysis is now part of the updated study in the “In vitro studies” section.

  1. Remodeling process time depends on the tissue: Please indicate the time scale for the remodeling of the tissues identified in the work (vascular grafts; bile ducts). Evaluation of the suitability of the scaffolds for tissue engineering purposes must take this time scale in consideration.

Remodeling time scale for vascular grafts can range from 2 to 12 months https://doi.org/10.1016/j.biomaterials.2018.08.063. Bile ducts range between 2 and 6 months doi: 10.3390/ma14237468. In the quoted study, DOI: 10.1002/jbm.b.34896, the biocompatibility and biodegradability of electrospun poly (1,4-butylene succinate) extended with 1,6-diisocyanatohexane scaffold was proved, with no physiological complications or rejection of the device and complete reabsorption in 120 days post-implant was found by NMR image analysis.

I believe that after amendments, considering the abovementioned issues, the manuscript may be published.

Reviewer 2 Report

The submitted article presents the fabrication of nanofibrous tubular scaffold by electrospinning of poly (1,4-butylene succinate) solutions. Scaffold morphology and mechanical properties were explored and compared to those of native conduits. The in vitro cytocompatibility of the obtained material was studied as well.

Specific points requiring attention that should be addressed by the authors are detailed below.

1.The Introduction should be more comprehensive including the known data in the literature up to know concerning the electrospinning of PBS.

2.The authors should underline why they choose to work with PBS and they do not use for instance other biodegradable polymers (PLA, PCL, etc.)

3.Define the molecular weight of the poly (1,4-butylene succinate).

4.Present the reaction for the synthesis of poly (1,4-butylene succinate) extended with 1,6-diisocyanatohexane.

5.Specify the producer of the high-voltage generator.

6.Specify the version of the ImageJ.

7.Specify the producer of the Bose testbench system.

8.Why do you use so low rotation speed – 40 rpm. It well known that higher speed results in better fiber orientation. Have you vary and studied the rotation speed?

9.The used distance is 15.5 cm. How have you determined exactly this one and not 15 or 16 cm for example?

10.The SEM analysis proved that the fiber diameter is around 1μm. These fibers are not nano. They are micro. So the title of the article should be changed.

11.Observing the presented in Figure 3 SEMs, the fibers seem with diameters even higher than 1 μm. The authors should present fiber distribution diagram.

12.Improve Figure 4B. The difference in the water droplets is hardly visible. Enlarge them and cut the unnecessary white space around them.

13.Explain the large error in residual mass of electrospun PBS after degradation in bile after 18 days?

14.Improve the images after DAPI staining (24 and 72 h) on Figure 7 A and B.

15.In the Section References: do not use et al. Use the template for the references of the Polymers.

Author Response

The submitted article presents the fabrication of nanofibrous tubular scaffold by electrospinning of poly (1,4-butylene succinate) solutions. Scaffold morphology and mechanical properties were explored and compared to those of native conduits. The in vitro cytocompatibility of the obtained material was studied as well.

Specific points requiring attention that should be addressed by the authors are detailed below.

  1. The Introduction should be more comprehensive including the known data in the literature up to know concerning the electrospinning of PBS.

Authors thank he reviewer for the suggestion; other studies on this topic are now discussed in the introduction of revised manuscript.

  1. The authors should underline why they choose to work with PBS and they do not use for instance other biodegradable polymers (PLA, PCL, etc.)

Accordingly, other explanations are now discussed in the introduction, lines 110-117 of revised manuscript.

  1. Define the molecular weight of the poly (1,4-butylene succinate).

The merchant Sigma Aldrich (CAS 143606-53-5 ) specifies only the Tm. But in this article http://dx.doi.org/10.1016/j.polymdegradstab.2014.03.033 (already cited at line 379 of revised manuscript) authors perform GPC analysis to evaluate the molecular weight (81.2kDa). The

  1. Present the reaction for the synthesis of poly (1,4-butylene succinate) extended with 1,6-diisocyanatohexane.

It is a commercial product, we bought it from Sigma Aldrich (CAS
143606-53-5). In this article http://dx.doi.org/10.1016/j.polymdegradstab.2014.03.033 (already cited at line 379 of revised manuscript) authors present the reaction for the synthesis of PBS prepolymer with hexamethylene diisocyanate (HDI).

  1. Specify the producer of the high-voltage generator.

The voltage generator is integrated in the NF 103 Electrospinning (producer MECC, Fukuoka, Japan).

  1. Specify the version of the ImageJ.

Thanks for the remark. ImageJ 1.52Q Wayne Rasband National Institute of Health, USA (now written in line 169).

  1. Specify the producer of the Bose testbench system.

Bose TA Instruments ElectroForce Test Bench System (now written in line 221 of revised manuscript).

  1. Why do you use so low rotation speed – 40 rpm. It well known that higher speed results in better fiber orientation. Have you vary and studied the rotation speed?

We have studied the rotation speed, and higher speed results in better fiber orientation indeed. Nevertheless, in this work, we want isotropic conducts therefore we have used a low rotation speed.

  1. The used distance is 15.5 cm. How have you determined exactly this one and not 15 or 16 cm for example?

In the present study we have optimized the electrospinning setup and processing variables to obtain a stable Taylor cone and a random fiber mat characterized by a homogeneous matrix with fiber diameters around 1um.

  1. The SEM analysis proved that the fiber diameter is around 1μ These fibers are not nano. They are micro. So the title of the article should be changed.

Accordingly to reviewer, the article title is now updated.

  1. Observing the presented in Figure 3 SEMs, the fibers seem with diameters even higher than 1 μ The authors should present fiber distribution diagram.

Fiber distribution diagram has been generated during the evaluation. The diagram was not reported in the manuscript. Actually, the information was added in the text of revised manuscript in line 320.

  1. Improve Figure 4B. The difference in the water droplets is hardly visible. Enlarge them and cut the unnecessary white space around them.

Thanks for the remark, the quality of image 4B was improved.

13.Explain the large error in residual mass of electrospun PBS after degradation in bile after 18 days?

The standard deviation after 18 days is 7%. The results after 18 days are exactly 0,00012g heavier or lighter than the original sample. So, this is probably an error attributable to the to the weighing operation.

14.Improve the images after DAPI staining (24 and 72 h) on Figure 7 A and B.

Accordingly, the quality of images 7 A and B was improved.

  1. In the Section References: do not use et al. Use the template for the references of the Polymers.

Thanks for the remark, the article is now updated.

Reviewer 3 Report

The manuscript explains some basic characteristic of electrospun PBS nanofibers. A list of concerns an suggestions:

1. The introduction must be improved, the paragraphs sound like disconnected from each other. The references must be enhanced (as an example for lines 42-43). The importance and challenge of preparing nanofibers in conduit shape must be addressed. 

2. Line 63 does not sound very scientific, can be replaced by %.

3. Line 76-78, english must be checked, use the noun forms instead of adjectives, such as "Designing requirements are hemocompatibility, elasticity, avoiding leakage, compatibility with the standard sutures, mechanically resistance,  sterilizability, reliabilility and appropriate length and diameter".

4. Better to check English of line 78-79 and add a reference.

5. Line 119 Tm m must be subscript

6. Line 125-128, ethics must be addressed for cell extraction from pigs and animals. 

7. Line 134, CO2 subscript

8. Line 146 SEM must be in brackets, and after this point scanning electron microscopy must be written as SEM.

9. Line 214, 1.5 mm squares is not clear, is it 1.5 mmor square shape of 1.5mmx1.5mm?

10. Line 213 scanning microscopy must be SEM, Line 259 only SEM.

11. The scale bars in SEM images are not readable, and do these 2 images belong to same sample? No need to put both of them, the magnified one can be put as an inset.

12. Reference needed for line 265-269.

13. Line 288, is not clear. what do the authors mean by "The resulting value of WVTR is three times higher than their highest value of permeability membrane which was considered extremely high??"

14. The water contact angles are very small in figure 4, they can not be read.

15. For 3.3, the degradation of the conduits must also be confirmed by SEM images.

16. Quality of figure 6b-c is low.

17. In 3.5, cell studies are not adequate. Cell viability and cytotoxicity must be shown. Image 7B is very dark and hard to observe. I suggest another staining method to indicate the nucleus and use of confocal microscopy. The proliferation of cells on the conduits must also be shown.

Author Response

The manuscript explains some basic characteristic of electrospun PBS nanofibers. A list of concerns an suggestions:

  1. The introduction must be improved, the paragraphs sound like disconnected from each other. The references must be enhanced (as an example for lines 42-43). The importance and challenge of preparing nanofibers in conduit shape must be addressed.

Thanks for the suggestion, other works on this topic are now discussed in the introduction.

  1. Line 63 does not sound very scientific, can be replaced by %.

Thanks for the remark; the manuscript is now updated in line 72 of revised manuscript.

  1. Line 76-78, english must be checked, use the noun forms instead of adjectives, such as "Designing requirements are hemocompatibility, elasticity, avoiding leakage, compatibility with the standard sutures, mechanically resistance, sterilizability, reliabilility and appropriate length and diameter".

Thanks for the remark, the article is now updated in the lines 79-81 of revised manuscript.

  1. Better to check English of line 78-79 and add a reference.

Accordingly, the manuscript is now updated (lines 81-83 of revised manuscript).

  1. Line 119 Tm m must be subscript

The correction was added in the revised version (line 129).

  1. Line 125-128, ethics must be addressed for cell extraction from pigs and animals.

During the study no cells were extracted from pigs or other animals. Phisiological fluids such as porcine bile, was donated from Istituto Zooprofilattico della Sicilia “A. Mirri,” Palermo, Italy, accordingly with European rules on animal experiments.

  1. Line 134, CO2 subscript

Thanks for the remark, the article is now updated (line 144).

  1. Line 146 SEM must be in brackets, and after this point scanning electron microscopy must be written as SEM.

Thanks for the remark, the article is now updated (line 164 of revised version).

  1. Line 214, 1.5 mm squares is not clear, is it 1.5 mm2 or square shape of 1.5mmx1.5mm?

Thanks for the remark, the article is now updated (line 245 of revised version).

  1. Line 213 scanning microscopy must be SEM, Line 259 only SEM.

Thanks for the remark, the article is now updated (lines 262 and 280 respectively).

  1. The scale bars in SEM images are not readable, and do these 2 images belong to same sample? No need to put both of them, the magnified one can be put as an inset.

Accordingly with reviewer suggestion, the article is now updated.

  1. Reference needed for line 265-269.

Thanks for the remark, the article is now updated.

  1. Line 288, is not clear. what do the authors mean by "The resulting value of WVTR is three times higher than their highest value of permeability membrane which was considered extremely high??"

In the quoted work authors consider an extremely high value of WVTR =4000; differently, our result is three times higher than their value.

  1. The water contact angles are very small in figure 4, they can not be read.

Thanks for the remark, the article is now updated.

  1. For 3.3, the degradation of the conduits must also be confirmed by SEM images.

According with reviewer suggestion, SEM of scaffold samples, after degradation studies, was carried out and images added in the revised manuscript. The analysis is now part of the updated study in the stability chapter.

  1. Quality of figure 6b-c is low.

Thanks for the remark, the figure is now updated.

  1. In 3.5, cell studies are not adequate. Cell viability and cytotoxicity must be shown. Image 7B is very dark and hard to observe. I suggest another staining method to indicate the nucleus and use of confocal microscopy. The proliferation of cells on the conduits must also be shown.

Accordingly, the quality of images 7 A and B was improved.

Live and dead staining of these samples resulted in worse quality of the images and results are even harder to observe. Confocal microscopy could really help in showing better images but unfortunately, we are not able to perform this analysis now.

The proliferation of cells on the conduits is really interesting to be analyzed, and we plan to evaluate it in the future. Indeed, to obtain this result we need to build a proper bioreactor to allow dynamic cell culture under peristaltic flow.

Round 2

Reviewer 1 Report

I think that the article is now suitable for publication.

Author Response

Authors thaks the reviewr for his positive evaluation

Reviewer 2 Report

After the made revisions the manuscript is improved and can be accepted for publication.

Author Response

Authors thank the reviewer for his positive reply.

Reviewer 3 Report

I thank the authors for the revisions. Some small suggestions are:

1. References should be added to introduction, lines 64-72, for the polymers used for porous scaffolds prepared by electrospinning and the mentioned prosthetic grafts.

2. You can include "accordingly with European rules on animal experiments" in the manuscript.

3.Scale bars in the added SEM images in Fig 5,7,8 are not visible, please make them visible.

4. In table 2: WVTR g/m2⋅24 h 25°, m2 must be superscript.

Author Response

Authors thank reviewer for the proposed suggestions and for his support to improve the quality of the manuscript. Accordingly, the manuscript was updated considering each point mentioned by reviewer.

Round 3

Reviewer 2 Report

The submitted manuscript has been improved and can be accepted for publication.

Author Response

(The authors gave the same response as above.)
